# Application of Artificial Intelligence in Measuring Novel pH-Impedance Metrics for Optimal Diagnosis of GERD

**DOI:** 10.3390/diagnostics13050960

**Published:** 2023-03-03

**Authors:** Ming-Wun Wong, Benjamin D. Rogers, Min-Xiang Liu, Wei-Yi Lei, Tso-Tsai Liu, Chih-Hsun Yi, Jui-Sheng Hung, Shu-Wei Liang, Chiu-Wang Tseng, Jen-Hung Wang, Ping-An Wu, Chien-Lin Chen

**Affiliations:** 1Department of Medicine, Hualien Tzu Chi Hospital, Buddhist Tzu Chi Medical Foundation, Tzu Chi University, 707, Section 3, Chung-Yang Road, Hualien 97004, Taiwan; 2School of Post-Baccalaureate Chinese Medicine, Tzu Chi University, Hualien 97004, Taiwan; 3Division of Gastroenterology, Hepatology and Nutrition, University of Louisville, Louisville, KY 40292, USA; 4Division of Gastroenterology, Washington University School of Medicine, St. Louis, MO 63110, USA; 5AI Innovation Research Center, Hualien Tzu Chi Hospital, Buddhist Tzu Chi Medical Foundation, Hualien 97004, Taiwan; 6NVIDIA AI Technology Center, NVIDIA Corporation, Taipei 11492, Taiwan; 7Department of Medical Research, Hualien Tzu Chi Hospital, Buddhist Tzu Chi Medical Foundation, Hualien 97004, Taiwan; 8Institute of Medical Sciences, Tzu Chi University, Hualien 97004, Taiwan

**Keywords:** artificial intelligence, gastroesophageal reflux disease, impedance-pH monitoring, reflux episodes, post-reflux swallow-induced peristaltic wave index

## Abstract

Novel metrics extracted from pH-impedance monitoring can augment the diagnosis of gastroesophageal reflux disease (GERD). Artificial intelligence (AI) is being widely used to improve the diagnostic capabilities of various diseases. In this review, we update the current literature regarding applications of artificial intelligence in measuring novel pH-impedance metrics. AI demonstrates high performance in the measurement of impedance metrics, including numbers of reflux episodes and post-reflux swallow-induced peristaltic wave index and, furthermore, extracts baseline impedance from the entire pH-impedance study. AI is expected to play a reliable role in facilitating measuring novel impedance metrics in patients with GERD in the near future.

## 1. Introduction

Gastro-esophageal reflux disease (GERD) is common, with an estimated global prevalence of 8–33% [1]. Upper endoscopy is frequently employed as the initial test in individuals with alarm symptoms or when symptoms fail to respond to antisecretory therapy. However, the sensitivity in identifying reflux disease by esophagogastroduodenoscopy is imperfect [2]. Where endoscopy fails to identify an etiology for patients’ symptoms, ambulatory reflux monitoring is performed to confirm or rule out pathological esophageal reflux burden. Additionally, newer clinical practice guidelines suggest that, for atypical symptoms, up-front reflux monitoring is a cost-saving approach that expedites personalized care [3]. Guidelines also advise its use ahead of foregut surgeries, particularly to document a relationship between reflux and symptoms ahead of anti-reflux surgery [4]. Newer evidence also suggests that those undergoing bariatric surgery may benefit from reflux monitoring to determine optimal surgical strategy [5,6,7].

The multitude of indications, as described above, in conjunction with the frequency of GERD, provide a framework for understanding the volume of ambulatory reflux monitoring studies performed each year [6]. Compared to pH-only evaluation, pH-impedance monitoring provides information about non-acid events, reflux episode numbers, and can describe the proximal extent of a reflux episode [8]. Additionally, for atypical disorders such as rumination syndrome or supra-gastric belching, the impedance component of ambulatory reflux monitoring allows the determining of directionality of esophageal events, and the contribution air events may play in symptoms [9]. Despite recent shifts in practice, pH impedance testing remains the standard for confirming or excluding pathological GERD, although not always conclusively [6].

Artificial intelligence (AI) is becoming widely used in different medical settings to improve the diagnosis and management of multiple medical conditions, including gastrointestinal (GI) diseases. The term “artificial intelligence” generically refers to complex computer algorithms that mimic human cognitive functions, including learning and problem-solving [10]. However, the term encompasses strategies that range from basic linear regressions to extremely complex neural networks designed to mirror human neural layering. Machine learning describes a subset of AI whereby the algorithm can be taught to discriminate characteristics of data samples and then apply experience to interpret previously unknown information [11]. Supervised machine learning is based on manual algorithms, in which the researcher manually indicates the features of interest in the input dataset (labeled dataset) to train the system to recognize discriminative features and provide appropriate output [10]. Deep learning is a subset of machine learning that can autonomously extract discriminative properties that identify input data through artificial neural networks, usually organized as convolutional neural networks (CNN) consisting of multiple layers of nonlinear functions [10,12,13].

In view of the recent increase in research investigating the use of artificial intelligence in clinical medicine, it is evident that practitioners will need to educate themselves regarding the basic underpinnings of the more common methodologies. Further, the field is fascinating and rapidly evolving, requiring thought and commitment to stay current. Accordingly, we reviewed the available evidence on the performance of AI in the field of the pH-impedance monitoring.

## 2. Machine Learning in Gastrointestinal Disease

AI refers to the ability of computers to exhibit human-like judgment or simulation. As technology advances, AI has evolved into various forms and applications, with early expert systems being one such example. However, these systems were rule-based and could lead to incorrect judgments due to poorly designed rules. Machine learning involves teaching the machine by quantifying the features of data, while deep learning uses a large amount of data and a multi-layer nonlinear structure neural network to allow the model to learn the features by itself. The more layers a deep model has, the stronger its learning ability, but a high number of layers can also result in degradation. The residual network is a breakthrough in deep models and is commonly used in current model architectures [14]. Table 1 summarizes the definition, application, advantages, and disadvantages of machine learning and deep learning.

Artificial intelligence is increasingly being used in gastroenterology, primarily for disease diagnosis and prognosis [13]. Breakthroughs in applications have come from the use of CNN to improve the training of these networks and build the foundation for artificial intelligence, as well as the development of powerful computing technology for advanced graphics processing. In the esophagus, artificial intelligence is currently used to detect Barrett’s esophagus with pathological low-grade dysplasia, assist endoscopy in identifying esophageal adenocarcinoma and squamous cell carcinoma and their invasion depth, and predict the effectiveness and prognosis of squamous cell carcinoma of the esophagus [15,16,17,18]. In the stomach, artificial intelligence can currently predict Helicobacter pylori infection, and assist endoscopy in identifying atrophy gastritis, small intestinal metaplasia, and early stomach cancer and their invasion depth and prognosis [19,20,21]. In the colon, artificial intelligence can currently assist endoscopy in diagnosing ulcerative colitis and Crohn’s disease and determining their disease severity, and in detecting colon polyps and lesion characteristics [13,22,23]. The underpinnings in image recognition are relatively familiar, even to the general public, given the iniquitousness of products—cars, home appliances, social media applications—that utilize some form of machine learning. However, it should be kept in mind, for the purposes of this article, that many of these technologies use preexisting neural networks [24]. That is to say that many publications use new data but employ established “machines” to do the learning. This is a decidedly positive factor. By the time many researchers get to a particular algorithm, the bugs, literally and figuratively, have been resolved.

However, pattern and image recognition are more complicated in non-traditional settings. For example, the interpretation of electrocardiogram tracings proved to be a more challenging task, as de novo methodologies were required and novel machine learning techniques need to be employed [25,26].

There are five important steps in the development of an AI model clinical implication:

A. Data collection and annotation: Gathering and labeling data sets for the AI model to learn from.

In order for an AI model to be effective, it needs to be trained on a large and diverse dataset that is representative of the patient population it will be used on. This means that the data should be collected from a variety of sources, and should include a diverse range of patients with different demographics and medical conditions. Additionally, the data needs to be annotated, or labeled, by medical experts. This is because the AI model uses the labels to learn the relationship between the data and the output. Without accurate labels, the model will not be able to accurately classify new images. This is why it is important to have medical experts annotate the data, as they have the knowledge and expertise to ensure that the data is labeled correctly. Annotation can be carried out manually or semi-automatically by medical experts who review the images and assign labels to them. This process can be time-consuming, but it is essential for training the model to make accurate predictions. Data annotation can also be performed using crowdsourcing platforms, which allows multiple annotators to work on the same image and increases the speed of annotation. In summary, high-quality, representative, and accurately labeled data is the foundation for any AI-based method, and without it the model’s performance will be limited.

B. Model development: Designing and training the AI model using the annotated data sets.

Model development is an important step in improving the performance of AI-based methods. One of the key techniques used in model development is deep learning, which involves training models with multiple layers of artificial neural networks. These layers are designed to extract features from the data and learn increasingly complex representations of the data as it is passed through the layers. This allows the model to learn more accurate representations of the data, which can lead to improved performance. Another important aspect of model development is the use of a large amount of data to train the models. This is because deep learning models have a large number of parameters and require a significant amount of data to effectively learn the underlying patterns and relationships in the data. The more data that is available to train the model, the more accurate the model can become. Additionally, model development also involves fine-tuning the hyperparameters of the model, such as number of layers, neurons, and learning rate. Model validation and testing are also important to evaluate the model’s performance. It is also important to use techniques such as cross-validation and regularization to prevent overfitting, which can occur when a model becomes too complex and is unable to generalize to new data. In summary, model development is a complex process that requires a combination of advanced machine learning techniques, large amounts of data, and careful tuning of the model’s hyperparameters. By using these techniques and following best practices, it is possible to improve the performance of AI-based methods and achieve better results.

C. Interpretability: Making the AI model’s decision-making process understandable to humans.

Interpretability is an important aspect of AI-based methods, particularly in the medical field, where it is important for the results to be understandable and actionable for medical experts. One way to make the results of AI-based methods more interpretable is by using explainable AI techniques, such as feature importance and decision tree visualization [27]. Feature importance is a technique that measures the relative contribution of each feature or input variable to the output of the model. This can provide insights into which features the model is using to make its predictions, and can help identify any features that may be redundant or not useful for the particular task. Another technique is decision tree visualization, which can provide a graphical representation of how the model is making its predictions. Decision trees are a type of model that can be used to represent the logic behind a prediction. They divide the input space into a set of regions and assign a label to each region. The final decision is based on the features of the input that fall into each region. This technique can help identify which features are driving the decisions of the model, and can provide insight into the reasoning behind the model’s predictions. It is worth noting that not all models are interpretable and explainable, e.g., black-box models such as deep neural networks. In those cases, techniques such as LIME [28], SHAP [29], and other methods that can provide explanations for specific predictions can be used. In summary, interpretability is an important aspect of AI-based methods and can be improved by using explainable AI techniques such as feature importance and decision tree visualization. These techniques can provide insights into how the model is making its predictions and can help medical experts understand and trust the results of the AI-based methods.

D. Clinical validation: Testing the AI model’s performance in a clinical setting.

Clinical validation is an important step in demonstrating the clinical utility and generalizability of AI-based methods in a real-world setting. Clinical validation studies involve testing the performance of the AI-based method on a diverse patient population under real-world conditions. This helps to determine how well the AI-based method performs in comparison to current standards of care and also, if it can be integrated into clinical workflows. These studies typically involve a large number of patients and often take place in multiple clinical sites to ensure that the results are generalizable to different patient populations and settings. The study design should be rigorously controlled, with the inclusion and exclusion criteria clearly defined, and a clear primary endpoint. It is also important to use appropriate evaluation metrics such as diagnostic accuracy, sensitivity, specificity and other metrics that are relevant to the specific task. These metrics can be used to evaluate the performance of the AI-based method against the standard of care, and to compare it to other AI-based methods that have been developed for the same task. Additionally, clinical validation studies provide an opportunity to assess the potential safety and ethical implications of using AI-based methods in a clinical setting. It is also necessary to ensure that the AI-based method is explainable, interpretable, and trustworthy. In summary, clinical validation is an essential step in demonstrating the clinical utility and generalizability of AI-based methods in a real-world setting. It should be conducted on a diverse patient population and use appropriate evaluation metrics, and also consider the potential safety, ethical, and trustworthiness implications of the AI-based method.

E. Collaboration: Working with healthcare professionals and other stakeholders to ensure the AI model aligns with their needs and benefits patients.

Collaboration allows for the development of AI-based methods that are tailored to the specific needs of the medical community and that can be integrated into clinical workflows. It also enables the development of methods that are sensitive to the specific characteristics of the patient population. Furthermore, it leads to the creation of an interpretable and explainable model, which is important for the acceptance and trust of the medical community. The collaboration also allows for the sharing of data and resources, which is essential for the development and validation of AI-based methods. Medical experts can provide access to large amounts of high-quality, annotated medical data, while computer scientists and engineers can provide the computational resources and expertise needed to train and test the AI-based methods. In addition, the collaboration can help to identify new research directions and opportunities, and to promote the dissemination of the developed methods to the medical community.

### 2.1. Acid Exposure Time (AET)

Acid exposure time (AET) is the primary parameter of pH impedance monitoring used to quantitatively measure reflux, which is calculated as the percentage of total measurement time during which the pH < 4 of the distal esophagus. AET is a well-reproducible parameter that can be accurately calculated automatically through analysis software, and pathological AET is a good predictor of the response of anti-reflux therapy [30,31,32]. The modern consensus of GERD diagnosis, Lyon Consensus, recommends that AET < 4% is considered physiological or normal, while AET > 6% is considered pathological. However, the interpretation of AET values between these two ranges is inconclusive and further investigation is needed [6].

### 2.2. The Number of Reflux Episodes and Reflux–Symptom Association

The number of reflux episodes, including acid, weakly acidic, or weakly alkaline, is the secondary parameter of pH impedance monitoring. The Lyon Consensus suggests that fewer than 40 reflux episodes within a 24-h period are considered physiological or normal, while more than 80 episodes are considered pathological. However, the interpretation of the number of reflux episodes between these two ranges is inconclusive [6]. The correlation between elevated reflux events and improvement with escalated anti-reflux therapy relies on limited data and requires future investigation [33].

More often used to make clinical decisions is reflux symptom association. Symptoms that occur within two minutes after a reflux episode are considered to be related to the reflux. The symptom index (SI) refers to the proportion of reflux episodes that are related to symptoms among all reflux episodes. A value of more than 50% is considered abnormal [34,35]. The symptom association probability (SAP) includes the presence or absence of symptoms and reflux episodes. Throughout the study period, statistical methods are used to determine whether symptoms and reflux episodes are associated [36,37]; if the probability of chance association (*p*-value) is <5%, this is considered to be affirmative. When the AET is indeterminate (i.e., between 4% and 6%), both the number of reflux episodes and the association of reflux symptoms are considered as adjunctive measures [4,6].

However, pH-impedance studies require time and operator experience to interpret accurately because automated analysis overcounts reflux events, especially non-acidic reflux events [38].

### 2.3. Baseline Impedance

Contact of reflux material with the esophageal submucosal sensory receptors may lead to reflux symptoms. The integrity of the esophageal mucosa is a protective barrier that is related to esophageal sensitivity [39,40,41]. Accumulated damage to the esophageal mucosa from reflux can cause dilated intercellular spaces, which correlates with the esophageal baseline impedance [42]. The mean nocturnal baseline impedance (MNBI) is the most commonly used method for measuring the baseline impedance from 24-h pH impedance monitoring, representing the longitudinal assessment of the esophageal mucosal integrity as a result of accumulated damage from reflux. This can supplement the limitations of AET, which may have daily variations due to its cross-sectional observation [43,44,45].

There are two methods of calculation; baseline impedance can be taken from the average of three 10-min periods from 1:00 to 3:00 p.m., or from the average of the entire nocturnal recumbent period [46,47]. Low baseline impedance values are positively correlated with both elevated AET and reflux symptom severity [48,49,50]. Healthy individuals, patients with treated GERD, and patients with functional esophageal disorders have higher baseline impedance values than those with erosive esophagitis and nonerosive reflux disease (NERD) [43,46,47,48,51]. Low MNBI can independently predict the efficacy of anti-reflux medical or surgical treatment [52], especially when AET is inconclusive [53].

### 2.4. Postreflux Swallow-Induced Peristaltic Wave (PSPW)

Refluxate in the distal esophagus can trigger primary peristaltic, which delivers salivary bicarbonate to reverse esophageal mucosal acidification, and epidermal growth factor, potentially to repair mucosal damage [54]. A swallow that occurs within 30 s of the occurrence of a reflux episode from 24-h pH impedance monitoring is called a post-reflux swallow-induced peristaltic wave (PSPW). The proportion of reflux episode in which PSPW occurs is called the PSPW index [55]. In early studies, a PSPW index threshold of 61% was used to distinguish between individuals with GERD and healthy controls [43]. Both MNBI and PSPW index can enhance the diagnostic value of 24-h pH impedance monitoring, particularly in distinguishing between NERD and functional heartburn [43,56,57]. However, analysis software often overestimates the number of reflux episodes, leading to the need for time-consuming manual identification of PSPW index, which limits its practicality.

### 2.5. Clinical Implications for pH-Impedance Conventional and Novel Metrics

Conventional metrics including AET, numbers of reflux episodes, and SAP can classify patients into NERD (AET > 6%), inconclusive GERD (4% ≤ AET ≤ 6%), reflux hypersensitivity (AET < 4%, positive SAP), and functional heartburn (AET < 4%, negative SAP) [6]. On the other hand, two novel metrics extracted from pH-impedance studies, MNBI and PSPW index, augment the diagnostic value of pH-impedance monitoring, especially when AET is inclusive or less pathologic [46,57]. Of all the metrics, only AET can rely on the original software calculation, while the others must be corrected by experienced expertise, especially numbers of reflux episodes and PSPW, which takes a lot of time to read the entire 24-h waveform. Therefore, artificial intelligence assistance is a new opportunity to simplify their application.

## 3. Application of Artificial Intelligence for pH-Impedance

### 3.1. Artificial Intelligence for Measuring the Number of Reflux Episodes

Identifying reflux episodes is the first step in interpreting impedance-based parameters, including reflux symptom association and PSPW index. In addition, an increase in the number of reflux episodes beyond 80 is considered to be supportive of GERD diagnosis [6]. Currently, reflux episodes are recommended to be selected first from analysis software [5], but analysis software often overestimates reflux episodes, especially non-acid events [55,58]. Furthermore, new reflux episodes may also be also manually identified from original commercialized software recognition [58,59]. Therefore, time-consuming manual checking of individual reflux episodes is required for accurate identification.

In the first published study, researchers manually identified 2049 impedance events from 24-h pH impedance monitoring for AI training. In this proof-of-concept study design, the artificial intelligence was comprised, primarily, of a complex decision tree. The model was generated using parameters automatically generated but initial steps were also hard-coded based on Wingate Consensus recommendations for pH impedance interpretation [38]. The Wingate consensus guidelines for reflux episodes state that mealtimes must be correctly identified and excluded before evaluating pH-impedance events. A reflux episode is defined as a 50% decrease in impedance that lasts for at least 4 s in at least two distal impedance channels, with retrograde propagation. Additionally, a pH decrease to less than 4.0 that occurs concurrently with a 4-s retrograde 50% impedance decrease after a belch episode is counted as a reflux episode. However, a pH decrease without impedance-detected reflux episode is counted as part of acid exposure time if it is not an artifact, but not as a reflux episode. The AI model used a decision tree analysis algorithm consisting of 24 nodes over nine layers. The AI model accurately identified 119 reflux episodes, 168 air events, and 1528 swallows and artifacts in the impedance events with an accuracy of 88.5%, demonstrating good sensitivity for AI in identifying impedance events [60]. The authors state that they hard coded the decision tree themselves, and that the algorithm they developed is proprietary. They did not use commercially available software such as Lexi or Alex net in their work. However, the authors did not provide further details about the specific steps involved in developing the decision tree, such as the criteria or parameters used for classification or the method of validation.

Another more recent study utilized 7939 impedance events from 106 patients who underwent pH-impedance for AI model establishment. The data collection for this study was obtained from consecutive patients aged 20 to 65 years with typical GERD symptoms (heartburn and/or acid regurgitation) for at least three months and negative endoscopy, who underwent 24-h impedance-pH monitoring without PPI therapy for at least seven days. In this study, the raw data for the AI program was obtained from commercial pH-impedance analysis software (Medical Measurement Systems, Enschede, Netherlands) in the form of comma-separated value (csv) files. Reflux events were preliminarily screened using pH-impedance study software and identified based on a time interval of 30 s prior to the event start time and 90 s after the event start time. The raw waveform signals were then transformed into RGB-colored images for a total time of 120 s using Python software. The AI program was developed using Python-based preprocessing and explored multiple AI and machine learning platforms. The raw data was converted into grayscale images for input into the deep residual learning model for image recognition, which was proposed for supervised learning based on human interpretation by three experts following the Wingate consensus. The model comprised 18 convolutional layers (ResNet18) and was trained to identify reflux episodes using the grayscale images. The data was separated into training and testing sets, with 90% of the impedance events randomly distributed to the training set and the remaining 10% to the testing set. It demonstrated that the accuracy of the AI model in recognizing reflux episodes was up to 87%, and in terms of the number of reflux episodes calculation for individual patient, the inter-rater agreement between the AI model and the manual annotation results was excellent (ICC = 0.965). The results showed the potential for reflux episodes to be accurately and efficiently identified by AI [61].

### 3.2. Artificial Intelligence for Measuring Baseline Impedance

A single study, cited above [60], has evaluated whether artificial intelligence can augment baseline impedance analysis for the prediction of response to therapy. Investigators identified a convenience sample of patients studied using combined pH and impedance testing both on and off therapy, as well as a group of asymptomatic volunteers. Novel, prototypical AI software, as described above, was utilized for the identification of esophageal events during testing. All events—reflux, air, and swallowing—were extracted throughout the duration of the study, leaving a “baseline” impedance value (artificial intelligence baseline impedance, AIBI) in both the upright and supine position that was free of artifactual interruptions. Additionally, a new metric, the upright:recumbent AIBI ratio (AIBI ratio) was investigated.

The investigators reported that recumbent AIBI and MNBI correlated; however, the upright AIBI did not. They reported similarly that, although MNBI, overall AIBI, and AIBI ratio off PPI were different between those with and without symptoms, only the AIBI ratio was different amongst management responders versus non-responders, defined by at least 50% symptom improvement, a diagnostic advantage over other metrics studied. Additionally, the AIBI ratio demonstrated an area under the curve on receiver operating characteristic analysis similar to AET (0.661 vs. 0.715, respectively) for the prediction of response to medical therapy. These findings, importantly, were not similar for individuals tested on PPI.

### 3.3. Artificial Intelligence for Measuring PSPW

After reflux occurs, a swallowing reflex is physiologically induced, called PSPW. This not only clears reflux material, but the saliva brought by swallowing can also neutralize stomach acid, which is a crucial anti-reflux mechanism [62,63]. The PSPW index can distinguish GERD from functional esophageal disorders and the responsiveness of PPI treatment, especially when AET is inconclusive [64,65]. However, the PSPW index cannot be automatically output by software and still relies on time-consuming manual recognition, which is an opportunity for AI to optimize the process.

Data were recently collected from consecutive patients aged 20 to 65 years old who had typical GERD symptoms (heartburn and/or acid regurgitation) for a minimum of three months and had a negative endoscopy result. These patients underwent 24-h impedance-pH monitoring while not taking PPI therapy for at least seven days. The study on the construction of the AI model involved 106 patients and 3761 reflux episodes. Initially, three experts annotated manually if PSPW appeared after reflux episodes, and the results were used to develop and verify the AI model through deep residual learning for image recognition with 18 convolutional layers. The model used a CNN learning method with a residual nets architecture and ResNet 18 as the model. The input data was transformed into images and the model was trained by learning the features of the data from a large amount of data, enabling it to recognize differences in the features and give answers. It demonstrated that the accuracy of the AI model in recognizing PSPW events was up to 82% and, in terms of the PSPW index calculation for individual patient, the inter-rater agreement between the artificial intelligence model and the manual annotation results was excellent (ICC = 0.921). The results showed the potential for PSPW to be accurately and efficiently identified by AI [61].

## 4. Future Perspectives

pH-impedance monitoring can provide confirmatory evidence of GERD, especially in patients with normal endoscopy. The most important metric is AET; however, when AET falls between 4% to 6%, it is considered inconclusive and other metrics are needed as evidence for a physiological diagnosis of GERD, which are summarized in Table 2. The number of reflux episodes and the PSPW index require manual identification, which is time-consuming, while MNBI only calculates the mucosal integrity during nighttime. We can expect to save time in the evaluation of the number of reflux episodes and PSPW index by using AI, and to obtain the mucosal impedance values for both daytime and nighttime, eliminating all noise and optimizing the precise diagnosis of GERD. To date, two pilot studies have explored the potential of AI to identify reflux episodes as well as PSPW, and created a novel parameter of baseline impedance-AIBI, which is summarized in Table 2 [60,61]. In essence, the most challenging aspect of GERD diagnosis is when endoscopy results are normal and the AET in the pH-impedance study falls within the inconclusive range of 4% to 6%. The first two published AI models can automate the assessment of physiological information such as the number of reflux episodes, MNBI, and PSPW index, which typically require time-consuming manual evaluation. It is important to note that AI is not intended to change the diagnostic process for GERD, but rather to provide complete physiological information in a more efficient manner. Further research will be required to validate AI potential in facilitating pH-impedance interpretation and to evaluate the impact in clinical practice. Of noted, collaboration between experts in computer science, engineering and medicine is crucial to fully leverage the potential of AI for GERD diagnosis. This allows for the development of AI-based methods that are tailored to the specific needs of the medical community, while ensuring that the models are interpretable, explainable, and trustworthy. Collaboration also enables the sharing of data and resources and the identification of new research directions and opportunities.

## 5. Limitation

Currently, there are still many limitations to the application of artificial intelligence in pH-impedance study. The training data is limited and needs to be expanded through federal training and validation using a large amount of data in the future to broaden its application. Moreover, the compatibility of AI models with different pH-impedance analysis software needs to be verified. Clinical validation studies on a diverse patient population are necessary to demonstrate the practical usefulness and broad applicability of AI-based methods in real-world settings. Furthermore, interpretability of AI-based methods is important, but currently this aspect has not been studied. Finally, the accuracy of AI-based methods also needs improvement, as it is currently less than 90%. Advances in AI model establishing techniques will help in its practical use in clinical applications.

## 6. Conclusions

In conclusion, current pilot research on AI for pH-impedance study is expected to measure impedance metrics accurately and efficiently. In addition, it can also measure novel parameters such as the PSPW index. In addition, AI can extract baseline impedance from the entire pH-impedance study by identifying and further eliminating various impedance events, which assists in predicting medical treatment effectiveness of GERD. With this, physicians can better utilize all the complementary metrics extracted from pH-impedance monitoring to supplement the limitations of AET, especially when AET falls in the ambiguous range of 4% to 6%. We believe that, in the near future, AI will be able to enter clinical practice and assist physicians in accurately and efficiently obtaining pH-impedance parameters, promoting precise physiological diagnosis of GERD.

## Figures and Tables

**Table 1 diagnostics-13-00960-t001:** The definition, application, advantages, and disadvantages for machine learning and deep learning.

	Machine Learning	Deep Learning
Definition	Learning the characteristics of data and finding the rules for its operation from data	Using multiple layers of nonlinear methods to learn data features
Application	Data clustering, anomaly value detection, search for optimal solutions	Image recognition, image style transfer, language translation, speech recognition
Advantages	Smaller model scale, model training and learning process is easier to understand	Widely applicable, able to solve complex problems
Disadvantages	Data with high dimensions or complexity is not easy to learn	Larger model size, usually requires a large amount of data training to achieve better results

**Table 2 diagnostics-13-00960-t002:** Metrics extracted from pH-impedance monitoring for GERD.

	Upper Limit of Physiologic Values	Role of AI
AET	4% >6%: pathologic 4–6%: inconclusive	N/A
Number of reflux episodes	40 >80: pathologic reflux; 40–80: inconclusive	Accuracy: 87–88.5%
MNBI	>2292 Ω	Advanced to artificial intelligence baseline impedance
PSPW index	>61%	Accuracy: 82%

GERD, gastro-esophageal reflux disease; AET, acid exposure time; MNBI, mean nocturnal baseline impedance; PSPW, post reflux swallow induced peristaltic wave, N/A, not applicable.

## Data Availability

Not applicable.

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
