# Peer review of "Application of Artificial Intelligence in Measuring Novel pH-Impedance Metrics for Optimal Diagnosis of GERD"

_diagnostics, 2023, doi:10.3390/diagnostics13050960_

Round 1

Reviewer 1 Report

This paper is focus on application of AI in GERD from pH-impedance monitoring parameters, the topic is very interesting. Several indexes related pH-impedance were introduced, and two related references on application of AI to distinguish GERD were also introduced. However, there is no information on how to use pH-impedance parameters to build the AI model or what to do with the AI method in the future. This is incomplete for a review paper. So, I think this paper should be improved before publication.

(1)   The content should be improved.

i.e., In the line123 ( Section 3.1), the value “… is less than 4” should add unit for the value.

In the Table.1, since residual network is one model of deep learning models, thus, the residual network should not be compared with that of machine learning and deep learning.

The author should enhance the second paragraph of section 2.

(2)   There are only two important references (Ref.57 and 58) related the topic. Can you introduce the recognition model in detail, such as what kind of parameters were fed in the AI model, and CNN network.

(3)   The topic is about AI model, however, from section 5 to 7, it is not easy for us to understand how to use pH-Impedance for the future AI model. Should we use the tradition index listed in Table.2 or combined them with AI model, such as XGBoost? Please enhance these sections.

Author Response

Dear the editors,

Thanks for your responses and giving us the opportunity for the revision regarding our manuscript, diagnostics-2153177, entitled “Application of Artificial Intelligence in Measuring Novel pH-Impedance”. We have made appropriate corrections and changes in points according to the reviewers’ suggestions. We have modified the text where appropriate. Our responses and changes are enclosed herewith. Due to our revised article exceeding 4500 words, can the type of article be changed from 'opinion' to 'review'? We look forward to hearing from you soon.

Best regards,

Dr Chien-Lin Chen, MD, PhD

Reviewer(s)' Comments to Author:
Reviewer: 1

Comments and Suggestions for Authors

This paper is focus on application of AI in GERD from pH-impedance monitoring parameters, the topic is very interesting. Several indexes related pH-impedance were introduced, and two related references on application of AI to distinguish GERD were also introduced. However, there is no information on how to use pH-impedance parameters to build the AI model or what to do with the AI method in the future. This is incomplete for a review paper. So, I think this paper should be improved before publication.

(1)   The content should be improved.

i.e., In the line123 ( Section 3.1), the value “… is less than 4” should add unit for the value.

Response:

Thanks for the comments. We have corrected it accordingly.

Note: line 235.

In the Table.1, since residual network is one model of deep learning models, thus, the residual network should not be compared with that of machine learning and deep learning.

Response:

Thanks for the comments. We have corrected the table 1 and rewritten the paragraph.

AI refers to the ability of computers to exhibit human-like judgment or simulation. As technology advances, AI has evolved into various forms and applications, with early expert systems being one such example. However, these systems were rule-based and could lead to incorrect judgments due to poorly designed rules. Machine learning in-volves teaching the machine by quantifying the features of data, while deep learning uses a large amount of data and a multi-layer nonlinear structure neural network to allow the model to learn the features on its own. The more layers a deep model has, the stronger its learning ability, but a high number of layers can also result in degradation. The residual network is a breakthrough in deep models and is commonly used in current model architectures. Table 1 summarizes the definition, application, advantages, and disadvantages of ma-chine learning and deep learning.

Note: line 78-102.

The author should enhance the second paragraph of section 2.

Response:

Thanks for the comments. We have enhanced the second paragraph of section 2.

There are five important steps in the development of an AI model clinical implication: A. Data collection and annotation: Gathering and labeling data sets for the AI model to learn from.

        In order for an AI model to be effective, it needs to be trained on a large and diverse dataset that is representative of the patient population it will be used on. This means that the data should be collected from a variety of sources, and should include a diverse range of patients with different demographics and medical conditions. Additionally, the data needs to be annotated, or labeled, by medical experts. This is because the AI model uses the labels to learn the relationship between the data and the output. Without accurate labels, the model will not be able to accurately classify new images. This is why it is important to have medical experts annotate the data, as they have the knowledge and expertise to ensure that the data is labeled correctly. Annotation can be done manually or semi-automatically by medical experts who review the images and assign labels to them. This process can be time-consuming, but it is essential for training the model to make accurate predictions. Data annotation can also be done using crowdsourcing platforms, which allows multiple annotators to work on the same image and increases the speed of annotation. In summary, high-quality, representative, and accurately labeled data is the foundation for any AI-based method, and without it, the model's performance will be limited.

  1. Model development: Designing and training the AI model using the annotated data sets.

Model development is an important step in improving the performance of AI-based methods. One of the key techniques used in model development is deep learning, which involves training models with multiple layers of artificial neural networks. These layers are designed to extract features from the data and learn increasingly complex representations of the data as it is passed through the layers. This allows the model to learn more accurate representations of the data, which can lead to improved performance. Another important aspect of model development is using a large amount of data to train the models. This is because deep learning models have a large number of parameters and require a significant amount of data to effectively learn the underlying patterns and relationships in the data. The more data that is available to train the model, the more ac-curate the model can become. Additionally, model development also involves fine-tuning the hyperparameters of the model such as number of layers, neurons, and learning rate. Model validation and testing are also important to evaluate the model's performance. It is also important to use techniques such as cross-validation and regularization to prevent overfitting, which can occur when a model becomes too complex and is unable to generalize to new data. In summary, model development is a complex process that requires a combination of advanced machine learning techniques, large amounts of data, and careful tuning of the model's hyperparameters. By using these techniques and following best practices, it is possible to improve the performance of AI-based methods and achieve better results.

  1. Interpretability: Making the AI model's decision-making process understandable to humans.

Interpretability is an important aspect of AI-based methods, particularly in the medical field where it is important for the results to be understandable and actionable for medical experts. One way to make the results of AI-based methods more interpretable is by using explainable AI techniques such as feature importance and decision tree visualization. Feature importance is a technique that measures the relative contribution of each feature or input variable to the output of the model. This can provide insights into which features the model is using to make its predictions, and can help identify any features that may be redundant or not useful for the particular task. Another technique is decision tree visualization, which can provide a graphical representation of how the model is making its predictions. Decision trees are a type of model that can be used to represent the logic behind a prediction. They divide the input space into a set of regions and assign a label to each region. The final decision is based on the features of the input that fall into each region. This technique can help identify which features are driving the decisions of the model, and can provide insight into the reasoning behind the model's predictions. It's worth noting that not all models are interpretable and explainable, such as black-box models like deep neural networks. In those cases, techniques such as LIME, SHAP, and other methods that can provide explanations for specific predictions can be used. In summary, interpretability is an important aspect of AI-based methods and can be im-proved by using explainable AI techniques such as feature importance and decision tree visualization. These techniques can provide insights into how the model is making its predictions and can help medical experts understand and trust the results of the AI-based methods.

  1. Clinical validation: Testing the AI model's performance in a clinical setting.

Clinical validation is an important step in demonstrating the clinical utility and generalizability of AI-based methods in a real-world setting. Clinical validation studies involve testing the performance of the AI-based method on a diverse patient population under real-world conditions. This helps to determine how well the AI-based method performs in comparison to current standard of care and also, if it can be integrated into clinical workflows. These studies typically involve a large number of patients and often take place in multiple clinical sites to ensure that the results are generalizable to different patient populations and settings. The study design should be rigorously controlled, with the inclusion and exclusion criteria clearly defined, and a clear primary endpoint. It's also important to use appropriate evaluation metrics such as diagnostic accuracy, sensitivity, specificity and other metrics that are relevant to the specific task. These metrics can be used to evaluate the performance of the AI-based method against the standard of care, and to compare it to other AI-based methods that have been developed for the same task. Additionally, clinical validation studies provide an opportunity to assess the potential safety and ethical implications of using AI-based methods in a clinical setting. It is also necessary to ensure that the AI-based method is explainable, interpretable, and trust-worthy. In summary, clinical validation is an essential step in demonstrating the clinical utility and generalizability of AI-based methods in a real-world setting. It should be conducted on a diverse patient population and use appropriate evaluation metrics, and also consider the potential safety, ethical, and trustworthiness implications of the AI-based method.

  1. Collaboration: Working with healthcare professionals and other stakeholders to en-sure the AI model aligns with their needs and benefits patients.

Collaboration allows for the development of AI-based methods that are tailored to the specific needs of the medical community and that can be integrated into clinical workflows. It also enables the development of methods that are sensitive to the specific characteristics of the patient population. Furthermore, it leads to the creation of an interpretable and explainable model, which is important for the acceptance and trust of the medical community. The collaboration also allows for the sharing of data and resources, which is essential for the development and validation of AI-based methods. Medical experts can provide access to large amounts of high-quality, annotated medical data, while computer scientists and engineers can provide the computational resources and expertise needed to train and test the AI-based methods. In addition, the collaboration can help to identify new research directions and opportunities, and to promote the dissemination of the developed methods to the medical community.

Note: line 129-230.

(2)   There are only two important references (Ref.57 and 58) related the topic. Can you introduce the recognition model in detail, such as what kind of parameters were fed in the AI model, and CNN network.

Response:

Thanks for the comments. Regarding Ref. 58. This research model adopts deep learning CNN and uses the ResNet 18 model. The input data is transformed into images and, through a large amount of data, the model learns the features and the differences in features to give the answer. We supplement this information in sections 4.1 and 4.3. Regarding reference 57, in this proof-of-concept study design, the artificial intelligence was comprised, primarily, of a complex decision tree. The model was generated using parameters automatically generated but also initial steps were hard-coded based on Wingate Consensus recommendations for pH impedance interpretation. This study's model had not yet adopted CNN but only used decision tree. We supplement this information in sections 4.1.

Section 4.1

In this proof-of-concept study design, the artificial intelligence was comprised, primarily, of a complex decision tree. The model was generated using parameters automatically generated but also initial steps were hard-coded based on Wingate Consensus recommendations for pH impedance interpretation.

The data collection for this study was obtained from consecutive patients aged 20 to 65 years with typical GERD symptoms (heartburn and/or acid regurgitation) for at least three months and negative endoscopy, who underwent 24-hour impedance-pH monitoring without PPI therapy for at least seven days. After manual classification of reflux and non-reflux events by three experts following the Wingate consensus, the results were utilized to develop and validate the AI model which employed a deep residual learning model for image recognition consisting of 18 convolutional layers. The model utilized in the study adopted the learning technique of CNN and had a residual nets architecture, using ResNet 18 as the model. The input data was transformed into images and the model was trained to learn the features of the data using a large amount of data, enabling it to identify differences in the features and give answers.

Note: line 315-319; line 325-338.

Section 4.3

A recent collected data from consecutive patients aged 20 to 65 years old who had typical GERD symptoms (heartburn and/or acid regurgitation) for a minimum of three months and had a negative endoscopy result. These patients underwent 24-hour impedance-pH monitoring while not taking PPI therapy for at least seven days.  The study on the construction of the AI model involved 106 patients and 3761 reflux episodes. Initially, three experts annotated manually if PSPW appeared after reflux episodes, and the results were used to develop and verify the AI model through deep residual learning for image recognition with 18 convolutional layers. The model used CNN learning method with a residual nets architecture and ResNet 18 as the model. The input data was transformed into images and the model was trained by learning the features of the data from a large amount of data, enabling it to recognize differences in the features and give answers.

Note: line 373-387.

(3)   The topic is about AI model, however, from section 5 to 7, it is not easy for us to understand how to use pH-Impedance for the future AI model. Should we use the tradition index listed in Table 2. or combined them with AI model, such as XGBoost? Please enhance these sections.

Response:

Thanks for the comments. The main tool for diagnosing GERD with pH-impedance monitoring is AET. Currently, AI models are effective and accurately calculating the diagnostic indicators for GERD with the aid of AET, especially when AET is inconclusive. With the help of AI, physicians can effectively use all the parameters of pH-impedance monitoring to make a precise diagnosis of GERD. We have rewritten sections 5 to 7 to make it easier for understanding.

  1. Future perspectives

The pH-impedance monitoring can provide confirmatory evidence of GERD, especially in patients with normal endoscopy. The most important metric is AET, however, when AET falls between 4% to 6%, it is considered inconclusive and other metrics are needed as evidence for a physiological diagnosis of GERD, which are summarized in Table 2. The number of reflux episodes and PSPW index require manual identification, which is time-consuming, while MNBI only calculates the mucosal integrity during nighttime. We can expect to save time in the evaluation of the number of reflux episodes and PSPW index by using AI, and to obtain the mucosal impedance values for both day-time and nighttime with AI eliminating all noise, optimizing the precise diagnosis of GERD. To date, two pilot studies have explored the potential of AI to identify reflux episodes as well as PSPW, and created a novel parameter of baseline impedance-AIBI, which was summarized in the Table 2. Further research will be required to validate AI potential in facilitating pH-impedance interpretation and to evaluate the impact in clinical practice. Of noted, collaboration between experts in computer science, engineering, and medicine is crucial to fully leverage the potential of AI for GERD diagnosis. It allows for the development of AI-based methods that are tailored to the specific needs of the medical community, while ensuring that the models are interpretable, explainable, and trustworthy. Collaboration also enables the sharing of data and resources and the identification of new research directions and opportunities.

Note: line 395-413.

  1. Limitation

Currently, there are still many limitations on the application of artificial intelligence on pH-impedance study. The training data is limited and needs to be expanded through federal training and validation using a large amount of data in the future to broaden its application. Moreover, the compatibility of AI models with different pH-impedance analysis software needs to be verified. Clinical validation studies on a diverse patient population are necessary to demonstrate the practical usefulness and broad applicability of AI-based methods in real-world settings. Furthermore, interpretability of AI-based methods is important, but currently, this aspect has not been studied. Finally, the accuracy of AI-based methods also needs improvement, as it is currently less than 90%. Advances in AI model establishing techniques will help in its practical use in clinical applications.

Note: line 417-434.

  1. Conclusion

In conclusion, current pilot research on AI for pH-impedance study is expected to accurately and efficiently measure impedance metrics. In addition, it can also measure the novel parameter such as PSPW index. In addition, AI can extract baseline impedance from the entire pH-impedance study by identifying and further eliminating various impedance events, which assists in predicting medical treatment effectiveness of GERD. With this, physicians can better utilize all the complementary metrics extracted from pH-impedance monitoring to supplement the limitations of AET, especially when AET falls in the ambiguous range of 4% to 6%. We believe that in the near future, AI will be able to enter clinical practice and assist physicians in accurately and efficiently obtaining pH-impedance parameters, promoting precise physiological diagnosis of GERD.

Note: line 435-446.

Reviewer 2 Report

This paper discusses the use of artificial intelligence (AI) in measuring novel metrics extracted from pH-impedance monitoring for the diagnosis of gastroesophageal reflux disease (GERD), which suggests that AI can accurately measure impedance metrics such as the number of reflux episodes and postreflux swallow-induced peristaltic wave index, and extract baseline impedance from pH-impedance studies. It is expected that AI will have a reliable role in measuring these novel impedance metrics in patients with GERD. The manuscript is of potential interest to the broad readers of Diagnostics.

However, there are some problems, which must be solved before it is considered for publication. If the following problems are well-addressed, this reviewer believes that the essential contribution of this paper is important for the diagnosis of GERD.

1. Data collection and annotation: To improve the accuracy of AI-based methods, it is important to collect high-quality data that is representative of the patient population. Additionally, it is important to have the data annotated by medical experts to ensure that the data is labeled correctly.

2. Model development: To improve the performance of AI-based methods, it is important to use advanced machine learning techniques such as deep learning. Additionally, it is important to use a large number of data to train the models.

3. Interpretability: To make the results of AI-based methods more interpretable, it is important to use explainable AI techniques such as feature importance and decision tree visualization.

4. Clinical validation: To demonstrate the clinical utility and generalizability of the AI-based methods in a real-world setting, it is important to conduct clinical validation studies in a diverse patient population.

5. Collaboration: To fully leverage the potential of AI for GERD diagnosis, collaboration between experts in computer science, engineering, and medicine should be encouraged.

Author Response

Dear the editors,

Thanks for your responses and giving us the opportunity for the revision regarding our manuscript, diagnostics-2153177, entitled “Application of Artificial Intelligence in Measuring Novel pH-Impedance”. We have made appropriate corrections and changes in points according to the reviewers’ suggestions. We have modified the text where appropriate. Our responses and changes are enclosed herewith. Due to our revised article exceeding 4500 words, can the type of article be changed from 'opinion' to 'review'? We look forward to hearing from you soon.

Best regards,

Dr Chien-Lin Chen, MD, PhD

Reviewer: 2

Comments and Suggestions for Authors

This paper discusses the use of artificial intelligence (AI) in measuring novel metrics extracted from pH-impedance monitoring for the diagnosis of gastroesophageal reflux disease (GERD), which suggests that AI can accurately measure impedance metrics such as the number of reflux episodes and postreflux swallow-induced peristaltic wave index, and extract baseline impedance from pH-impedance studies. It is expected that AI will have a reliable role in measuring these novel impedance metrics in patients with GERD. The manuscript is of potential interest to the broad readers of Diagnostics.

However, there are some problems, which must be solved before it is considered for publication. If the following problems are well-addressed, this reviewer believes that the essential contribution of this paper is important for the diagnosis of GERD.

  1. Data collection and annotation: To improve the accuracy of AI-based methods, it is important to collect high-quality data that is representative of the patient population. Additionally, it is important to have the data annotated by medical experts to ensure that the data is labeled correctly.

Response:

Thanks for the comments. Because your reminders are too important, we have written this content into section 2 and revised section 4.1 and 4.3.

Section 2

There are five important steps in the development of an AI model clinical implication: A. Data collection and annotation: Gathering and labeling data sets for the AI model to learn from.

        In order for an AI model to be effective, it needs to be trained on a large and diverse dataset that is representative of the patient population it will be used on. This means that the data should be collected from a variety of sources, and should include a diverse range of patients with different demographics and medical conditions. Additionally, the data needs to be annotated, or labeled, by medical experts. This is because the AI model uses the labels to learn the relationship between the data and the output. Without accurate labels, the model will not be able to accurately classify new images. This is why it is important to have medical experts annotate the data, as they have the knowledge and expertise to ensure that the data is labeled correctly. Annotation can be done manually or semi-automatically by medical experts who review the images and assign labels to them. This process can be time-consuming, but it is essential for training the model to make accurate predictions. Data annotation can also be done using crowdsourcing platforms, which allows multiple annotators to work on the same image and increases the speed of annotation. In summary, high-quality, representative, and accurately labeled data is the foundation for any AI-based method, and without it, the model's performance will be limited.

Note: line 129-146.

Section 4.1

The data collection for this study was obtained from consecutive patients aged 20 to 65 years with typical GERD symptoms (heartburn and/or acid regurgitation) for at least three months and negative endoscopy, who underwent 24-hour impedance-pH monitoring without PPI therapy for at least seven days.

Note: line 325-329.

Section 4.3

A recent collected data from consecutive patients aged 20 to 65 years old who had typical GERD symptoms (heartburn and/or acid regurgitation) for a minimum of three months and had a negative endoscopy result. These patients underwent 24-hour impedance-pH monitoring while not taking PPI therapy for at least seven days.  The study on the construction of the AI model involved 106 patients and 3761 reflux episodes.

Note: line 373-376.

  1. Model development: To improve the performance of AI-based methods, it is important to use advanced machine learning techniques such as deep learning. Additionally, it is important to use a large number of data to train the models.

 Response:

Thanks for the comments. Because your reminders are too important, we have written this content into section 2 and revised section 4.1, 4.3 and limitations.

Section 2.

  1. Model development: Designing and training the AI model using the annotated data sets.

Model development is an important step in improving the performance of AI-based methods. One of the key techniques used in model development is deep learning, which involves training models with multiple layers of artificial neural networks. These layers are designed to extract features from the data and learn increasingly complex representations of the data as it is passed through the layers. This allows the model to learn more accurate representations of the data, which can lead to improved performance. Another important aspect of model development is using a large amount of data to train the models. This is because deep learning models have a large number of parameters and require a significant amount of data to effectively learn the underlying patterns and re-lationships in the data. The more data that is available to train the model, the more ac-curate the model can become. Additionally, model development also involves fine-tuning the hyperparameters of the model such as number of layers, neurons, and learning rate. Model validation and testing are also important to evaluate the model's performance. It is also important to use techniques such as cross-validation and regularization to prevent overfitting, which can occur when a model becomes too complex and is unable to generalize to new data. In summary, model development is a complex process that requires a combination of advanced machine learning techniques, large amounts of data, and careful tuning of the model's hyperparameters. By using these techniques and following best practices, it is possible to improve the performance of AI-based methods and achieve better results.

Note: line 148-168.

Section 4.1

After manual classification of reflux and non-reflux events by three experts following the Wingate consensus, the results were utilized to develop and validate the AI model which employed a deep residual learning model for image recognition consisting of 18 convolutional layers. The model utilized in the study adopted the learning technique of CNN and had a residual nets architecture, using ResNet 18 as the model. The input data was transformed into images and the model was trained to learn the features of the data using a large amount of data, enabling it to identify differences in the features and give answers.

Note: line 329-338.

Section 4.3

The study on the construction of the AI model involved 106 patients and 3761 reflux episodes. Initially, three experts annotated manually if PSPW appeared after reflux episodes, and the results were used to develop and verify the AI model through deep residual learning for image recognition with 18 convolutional layers. The model used CNN learning method with a residual nets architecture and ResNet 18 as the model. The input data was transformed into images and the model was trained by learning the features of the data from a large amount of data, enabling it to recognize differences in the features and give answers.

Note: line 376-387.

Limitation

The training data is limited and needs to be expanded through federal training and validation using a large amount of data in the future to broaden its application.

Note: line 420-422.

  1. Interpretability: To make the results of AI-based methods more interpretable, it is important to use explainable AI techniques such as feature importance and decision tree visualization.

 Response:

Thanks for the comments. Because your reminders are too important, we have written this content into section 2 and revised limitations.

Section 2

  1. Interpretability: Making the AI model's decision-making process understandable to humans.

Interpretability is an important aspect of AI-based methods, particularly in the medical field where it is important for the results to be understandable and actionable for medical experts. One way to make the results of AI-based methods more interpretable is by using explainable AI techniques such as feature importance and decision tree visualization. Feature importance is a technique that measures the relative contribution of each feature or input variable to the output of the model. This can provide insights into which features the model is using to make its predictions, and can help identify any features that may be redundant or not useful for the particular task. Another technique is decision tree visualization, which can provide a graphical representation of how the model is making its predictions. Decision trees are a type of model that can be used to represent the logic behind a prediction. They divide the input space into a set of regions and assign a label to each region. The final decision is based on the features of the input that fall into each region. This technique can help identify which features are driving the decisions of the model, and can provide insight into the reasoning behind the model's predictions. It's worth noting that not all models are interpretable and explainable, such as black-box models like deep neural networks. In those cases, techniques such as LIME, SHAP, and other methods that can provide explanations for specific predictions can be used. In summary, interpretability is an important aspect of AI-based methods and can be im-proved by using explainable AI techniques such as feature importance and decision tree visualization. These techniques can provide insights into how the model is making its predictions and can help medical experts understand and trust the results of the AI-based methods.

Note: line 170-193.

Limitations

Furthermore, interpretability of AI-based methods is important, but currently, this aspect has not been studied.

Note: line 425-426.

  1. Clinical validation: To demonstrate the clinical utility and generalizability of the AI-based methods in a real-world setting, it is important to conduct clinical validation studies in a diverse patient population.

 Response:

Thanks for the comments. Because your reminders are too important, we have written this content into section 2 and revised limitations.

Section 2

  1. Clinical validation: Testing the AI model's performance in a clinical setting.

Clinical validation is an important step in demonstrating the clinical utility and generalizability of AI-based methods in a real-world setting. Clinical validation studies involve testing the performance of the AI-based method on a diverse patient population under real-world conditions. This helps to determine how well the AI-based method performs in comparison to current standard of care and also, if it can be integrated into clinical workflows. These studies typically involve a large number of patients and often take place in multiple clinical sites to ensure that the results are generalizable to different patient populations and settings. The study design should be rigorously controlled, with the inclusion and exclusion criteria clearly defined, and a clear primary endpoint. It's also important to use appropriate evaluation metrics such as diagnostic accuracy, sensitivity, specificity and other metrics that are relevant to the specific task. These metrics can be used to evaluate the performance of the AI-based method against the standard of care, and to compare it to other AI-based methods that have been developed for the same task. Additionally, clinical validation studies provide an opportunity to assess the potential safety and ethical implications of using AI-based methods in a clinical setting. It is also necessary to ensure that the AI-based method is explainable, interpretable, and trust-worthy. In summary, clinical validation is an essential step in demonstrating the clinical utility and generalizability of AI-based methods in a real-world setting. It should be conducted on a diverse patient population and use appropriate evaluation metrics, and also consider the potential safety, ethical, and trustworthiness implications of the AI-based method.

Note: line 195-215.

Limitations

The training data is limited and needs to be expanded through federal training and validation using a large amount of data in the future to broaden its application. Moreover, the compatibility of AI models with different pH-impedance analysis software needs to be verified. Clinical validation studies on a diverse patient population are necessary to demonstrate the practical usefulness and broad applicability of AI-based methods in real-world settings.

Note: line 420-425.

  1. Collaboration: To fully leverage the potential of AI for GERD diagnosis, collaboration between experts in computer science, engineering, and medicine should be encouraged.

Response:

Thanks for the comments. Because your reminders are too important, we have written this content into section 2 and revised section 5.

Section 2

  1. Collaboration: Working with healthcare professionals and other stakeholders to en-sure the AI model aligns with their needs and benefits patients.

Collaboration allows for the development of AI-based methods that are tailored to the specific needs of the medical community and that can be integrated into clinical workflows. It also enables the development of methods that are sensitive to the specific characteristics of the patient population. Furthermore, it leads to the creation of an interpretable and explainable model, which is important for the acceptance and trust of the medical community. The collaboration also allows for the sharing of data and resources, which is essential for the development and validation of AI-based methods. Medical experts can provide access to large amounts of high-quality, annotated medical data, while computer scientists and engineers can provide the computational resources and expertise needed to train and test the AI-based methods. In addition, the collaboration can help to identify new research directions and opportunities, and to promote the dissemination of the developed methods to the medical community.

Note: line 217-230.

Section 5

Of noted, collaboration between experts in computer science, engineering, and medicine is crucial to fully leverage the potential of AI for GERD diagnosis. It allows for the development of AI-based methods that are tailored to the specific needs of the medical com-munity, while ensuring that the models are interpretable, explainable, and trustworthy. Collaboration also enables the sharing of data and resources and the identification of new research directions and opportunities.

Note: line 408-413.

Round 2

Reviewer 1 Report

The authors have addressed most of the questions, and improved the paper in logicality and completeness. However, many details in this manuscript are not specified, especially, how did others use AI in GRED. It still needs to be improved before publication.

Comments on the authors are as follows:

(1) The name of Table 1 is not suitable, maybe compared between machine learning and deep learning.

(2) Concerned our previous question(Ref.57 and 58 or In Section 4.1), what kind of parameters were fed in the AI model and CNN network?There is no clear and fundamental answer to this question. What are the features of GERD? How did the author transform them into images for CNN model?

(3) In section 5, please present or enhance how to combine deep learning or machine learning methods in GRED?

Author Response

Comments on the authors are as follows:

(1) The name of Table 1 is not suitable, maybe compared between machine learning and deep learning.

Response:

Thanks for the comment. We have revised the name of Table 1 into “The definition, application, advantages, and disadvantages for machine learning and deep learning”.

(2) Concerned our previous question(Ref.57 and 58 or In Section 4.1), what kind of parameters were fed in the AI model and CNN network?There is no clear and fundamental answer to this question. What are the features of GERD? How did the author transform them into images for CNN model?

Response:

Thanks for the comments. It was suggested that an increase in the number of reflux episodes beyond 80 can assist in the diagnosis of GERD.

In reference 57, the AI model utilized a decision tree analysis algorithm and did not employ CNN. We have revised it accordingly. The authors developed the decision tree algorithm themselves and it is proprietary, with no use of commercially available software such as Lexi or Alex net. However, the authors did not provide detailed information regarding the specific steps involved in developing the decision tree, such as the criteria or parameters used for classification, or the method of validation.

In reference 58, we revised it as following: In this study, the raw data for the AI program was obtained from a commercial pH-impedance analysis software (Medical Measurement Systems, Enschede, Netherlands) in the form of comma-separated value (csv) files. Reflux events were preliminarily screened using pH-impedance study software and identified based on a time interval of 30 seconds prior to the event start time and 90 seconds after the event start time. The raw waveform signals were then transformed into RGB-colored images for a total time of 120 seconds using Python software. The AI program was developed using Python-based preprocessing and explored multiple AI and machine learning platforms. The raw data was converted into grayscale images for input into the deep residual learning model for image recognition, which was proposed for supervised learning based on human inter-pretation by three experts following the Wingate consensus. The model comprised 18 convolutional layers (ResNet18) and was trained to identify reflux episodes using the grayscale images. The data was separated into training and testing sets, with 90% of the impedance events randomly distributed to the training set and the remaining 10% to the testing set.

Note: line 295-297; line 307-314; line 315; line 318-322; line 328-341.

(3) In section 5, please present or enhance how to combine deep learning or machine learning methods in GRED?

Response:

Thanks for the comment. We have revised the section 5:

In essence, the most challenging aspect of GERD diagnosis is when endoscopy results are normal and the AET in pH-impedance study falls within the inconclusive range of 4% to 6%. The first two published AI models can automate the assessment of physiological information such as the number of reflux episodes, MNBI, and PSPW index, which typically require time-consuming manual evaluation. It is important to note that AI is not intended to change the diagnostic process for GERD, but rather to provide complete physiological information in a more efficient manner.

Note: line 412-418.

Reviewer 2 Report

Accept in present form

Author Response

Thank you for taking the time to review and provide thoughtful suggestions. Your expertise and insights are greatly appreciated.

Round 3

Reviewer 1 Report

The authors had addressed all the questions.